# Unusual Inflammatory Tinea Infections: Majocchi’s Granuloma and Deep/Systemic Dermatophytosis

**DOI:** 10.3390/jof7110929

**Published:** 2021-10-31

**Authors:** Jade Castellanos, Andrea Guillén-Flórez, Adriana Valencia-Herrera, Mirna Toledo-Bahena, Erika Ramírez-Cortés, Sonia Toussaint-Caire, Carlos Mena-Cedillos, Marcela Salazar-García, Alexandro Bonifaz

**Affiliations:** 1Dermatology Department, Hospital Infantil de México Federico Gómez, Mexico City 06720, Mexico; djade.castellanos@gmail.com (J.C.); andiie.guillen@gmail.com (A.G.-F.); mirnatoledo@gmail.com (M.T.-B.); camenac@gmail.com (C.M.-C.); 2Star Médica Hospital Infantil Privado, Mexico City 03810, Mexico; eggyst@hotmail.com; 3Dermatology Division, Hospital General Dr. Manuel Gea González, Mexico City 14080, Mexico; reportestouss@gmail.com; 4Biomedical Research Department, Hospital Infantil de México Federico Gómez, Mexico City 06720, Mexico; marcelasalazargarcia@hotmail.com; 5Dermatology & Mycology Service, Hospital General de México Dr. Eduardo Liceaga, Mexico City 06720, Mexico; a_bonifaz@yahoo.com.mx

**Keywords:** inflammatory tinea, Majocchi’s granuloma, dermatophytic, Hadida

## Abstract

Purpose of review. Inflammatory tinea is an uncommon group of dermatophyte entities that predominantly cause fungal infection of the skin and hair. This review intends to present all of the available evidence regarding its epidemiology, etiopathogenesis, clinical features, and diagnostic methods as well as treatments recommended for various inflammatory tinea infections. This article provides a review of Majocchi’s granuloma and dermatophytic or Hadida’s disease. Recent findings. The new phylogenetic classification of dermatophytes includes nine genera, and those that affect humans are *Trichophyton*, *Microsporum*, *Epidermophyton*, and *Nannizzia*. Furthermore, molecular advancements have revealed impaired antifungal immune responses caused by inflammatory tinea, which are detailed in this article. Summary. The common denominator in these pathologies is the presence of impaired immune responses and, consequently, an impaired inflammatory response by the host. It is necessary to be familiar with these immunological characteristics in order to use the appropriate diagnostic methods and to provide adequate treatment.

## 1. Introduction

Dermatophytes are fungi that invade and multiply within keratinized tissues (skin, hair, and nails), causing infection. The new phylogenetic classification of dermatophytes includes nine genre, and those that affect humans are *Trichophyton*, *Microsporum*, *Epidermophyton,* and *Nannizzia*. Finally, based upon the affected site, these have been classified clinically into tinea capitis, tinea faciei, tinea barbae, tinea corporis, tinea manus, tinea cruris, tinea pedis, and tinea unguium.

Other unusual clinical variants include Majocchi granuloma and Dermatophytic disease, and although usually painless and superficial, these fungi can behave in an invasive manner, causing deeper and disseminated infection and should not be neglected.

Majocchi’s granuloma represents a rare, deep dermatophyte folliculitis; it is considered a clinical variant of tinea corporis, which is usually caused by the dermatophyte *T. rubrum*. Two main clinical forms have been described: papulopustular perifollicular and indurated plaques with erythematous subcutaneous nodules. The former is typically developed in trauma-prone areas in healthy individuals, whereas the latter presents in immunosuppressed patients.

Dermatophytic disease is a chronic dermatophyte infection of the skin and viscera and mainly occurs in Africa. Immunological studies have highlighted a cellular immunity deficit, with autosomal recessive transmission being responsible for tolerance to dermatophytes. The first signs of this disease usually occur during childhood; it is a serious life-threatening disease due to its inexorable evolution toward visceral involvement. The present review is an important topic and will explore the recent advances in pathophysiology and management.

## 2. Majocchi’s Granuloma

### 2.1. Definition

Majocchi’s granuloma (MG), also known as Majocchi’s trichophytic granuloma, dermatophytic granuloma, or nodular granulomatous perifolliculitis, is a skin infection of the dermal and/or subcutaneous tissue that is caused by dermatophytes; the most commonly used term is Majocchi’s granuloma [1,2]. It was first described in 1883 in Italy by Professor Domenico Majocchi as a granulomatous inflammatory process caused by the local invasion of a dermatophyte, which he called “trichophytic granuloma” [3,4].

### 2.2. Etiology

The etiology of MG changes according to the geographic region. *T. rubrum* is the most frequently isolated microorganism in immunocompetent and immunocompromised individuals [5,6,7,8,9,10,11]. However, other pathogens have also been described: *T. mentagrophytes*, *T. tonsurans*, *T. verrucosum*, *M. canis*, and *E. floccosum*. Smith et al. identified the following dermatophyte fungi in patients with MG: *T. violaceum*, *M. audouinii*, *Nannizzia gypsea*, and *M. ferrugineum* [4,5].

### 2.3. Epidemiology

The disease is of a cosmopolitan nature, predominantly occurring in the age groups between 20 and 35 years; generalized MG cases have been described in children aged 3–5 years of age [4]. It is predominant among females in a 3:1 ratio because they are more vulnerable to developing tinea capitis after puberty and because they generally shave their legs. MG cases among men are associated with immunosuppression [4].

The predisposing factors include long-term steroid use, chemotherapy, antineoplastic therapy, malnutrition, leukemia, lymphoma, acquired immunodeficiency syndrome (AIDS), and Cushing’s syndrome [12]. A triggering factor is physical trauma, such as that caused by shaving of the legs and the blockage of the hair follicles, which directly or indirectly leads to the rupture of the follicle and a passive introduction of the microorganism [6].

### 2.4. Pathogeny

Dermatophytes degrade keratin in non-living keratinized tissues to survive. In the case of MG, the fungi must survive in the dermal and subcutaneous tissues. In particular, the dermal environment is more alkaline than that of the epidermis; therefore, the dermis does not have ideal conditions for microorganism growth [13].

Cell destruction associated with fungal growth as well as the increase in the amounts of stromal acid mucopolysaccharides produced by inflammation reduces dermal pH, generating a favorable environment for the survival of the fungal pathogen [5,13].

There are some theories on pathogenic mechanisms that are based on factors associated with the host and microorganism [14]:The physical skin barrier prevents the entry of microorganisms [15,16,17].Antimicrobial peptides (cathelicidins) protect the skin against dermatophytes and help limit their dermal invasion [18,19,20].Nonspecific phagocytic functions of neutrophils and macrophages are crucial in the control of deep or invasive fungal infections [9].

### 2.5. Clinical Features

It is a localized infection that most frequently affects lower extremities in the adult population and the head in the case of pediatric patients. It is characterized by small violaceous papules and nodules or plaques and has three clinical phases from the morphological point of view: plaques, nodules, and degeneration [4,6,7].

On hairy skin, the first phase may be similar to a dry tinea of the head, with erythematous-squamous plaques and short hairs. Next, the second phase is characterized by red-purple nodules of up to 2 cm in diameter, which are painful upon palpation and have a tendency to break outward. Ulcers are the primary manifestation of the degenerative phase [4].

On non-hairy skin, the first phase is similar to the body ringworm or *Tinea corporis*, with the formation of very pruritic, erythematous-squamous plaques with erythematous borders. In the second phase, nodules of up to 3 cm tend to cluster around the plate, giving the appearance of a “nodosum cord,” and ultimately, these lesions ulcerate [1,3,7,14] (Figure 1A).

There are reports of deep nodular MG due to immunosuppression associated with the use of standalone or combined medications, such as prednisone, vincristine, cyclophosphamide, or azathioprine. Radentz et al. [12] reported on the skin of six patients with multiple fluid-filled, dark red to purple lesions that were tender to touch as well as with ulcerated nodules, papules, and lesions suggesting vasculitis. All patients had a history of chronic dermatophytosis [12].

MG in genitalia is an unusual presentation, and to date, only two cases have been reported. Chang et al. reported a vulvar lesion caused by *T. mentagrophytes* in a 23-year-old woman with chronic eczema. The patient had been using topical steroids for 5 years, her dog was the most likely source of infection [21]. Cho et al. reported a case with superficial peripheral follicular involvement caused by *T. rubrum* in a 66-year-old man. Dermatosis was found in the scrotal skin; the patient was healthy with a history of jock itch (tinea cruris) [22].

### 2.6. Diagnosis

KOH exam is insufficient to distinguish superficial and invasive dermatophytosis because peripheral granulomatous inflammation must be indicated histologically for the diagnosis of MG [14].

Identifying the agent by culture is essential; it is recommended to culture the exudate collected from ulcers or to obtain the sample from a nodule or using fine-needle aspiration [3,4].

The intradermal reaction to trichophyton is negative [4].

### 2.7. Histopathology

MG should be diagnosed using a histopathological examination and is based on the detection of granulomas in the mid and deep dermis [5]. Staining methods, such as periodic acid-Schiff (PAS) or Gomori methenamine silver (GMS), can be implemented [8].

Histopathological analysis reveals suppurative and peripheral folliculitis with fungal hyphae in the affected hair follicles associated with perifollicular granulomatous inflammation and giant cell reaction caused by the fungal elements [8] (Figure 1B,C).

Extensive tissue necrosis and abscess formation with less extensive epidermal acanthosis and a less granulomatous reaction are characteristic features of samples from immunocompromised patients [5].

### 2.8. Differential Diagnosis

This should be performed for diseases involving chronic erythematous papules and nodules [3,9]. These include mycobacterial infections, deep fungal infections, disseminated toxoplasmosis, and cutaneous leishmaniasis [9,14].

When the face is affected, granulomatous rosacea and granuloma faciale should be considered. Painful nodules resemble erythema nodosum, thrombophlebitis, and erythema induratum of Bazin. In immunosuppressed patients, it should be distinguished from Kaposi’s sarcoma and lymphoma [14]. MG affecting hairy skin should be differentiated from folliculitis, tinea capitis, and kerion celsi [23].

### 2.9. Prognosis

Recurrence is expected because the foci of dermatophytosis remain uncured in nails, feet, or other anatomical sites [21]. An important treatment principle is to reduce immunosuppression by adjusting the doses of immunosuppressant agents [8]. Post-inflammatory pigmentation, atrophic scarring, and alopecia may develop after antifungal therapy [14].

### 2.10. Treatment

The best therapeutic option is terbinafine (250–500 mg/day) for adults, which has fungistatic action through the inhibition of ergosterol synthesis (by the blockade of squalene epoxidase) and is fungicidal, particularly against dermatophytes (by accumulation of squalene in a proximal step before squalene epoxidase) [24]. In addition to its increased efficacy in the elimination of dermatophytes, it poses a low risk of drug interactions and can be detected in the stratum corneum 24 h after administration [9]. Treatment duration should be 1–6 months and should be continued until the lesions are completely healed [2,4]. Gastrointestinal side effects are frequent, and include (5%) dyspepsia, retching, diarrhea, and abdominal pain as well as taste disturbances (2%) such as dysgeusia and reversible taste loss that can last up to 2 months. Exanthema and urticaria, appetite loss, headache and visual disturbances (blurry vision, visual deficiency and decreased visual acuity) arthralgias, myalgias, depression, and fatigue have all been described. Serious side effects are mainly dermatological and include issues such as the exacerbation of eczema, of psoriasis, or lupus erythematosus, or more seriously, acute exanthematic pustulosis and drug reactions with esosinofilia as well as systemic symptoms, such as Stevens–Jhonson syndrome and toxic epidermal necrolysis [24].

Determining liver function is recommended before prescribing terbinafine, and this should be repeated 4–6 weeks after using the drug or if the clinical symptomatology suggests liver involvement. An elevation of liver enzymes justifies the immediate interruption of terbinafine. However, this phenomenon is rare. In most cases, severe liver failure has been described in patients with a severe underlying condition. Some cases of neutropenia, thrombocytopenia, agranulocytosis, and pancytopenia reported during the use of terbinafine may justify a monitoring of the complete blood count [24].

Other options are itraconazole (200 mg/day), voriconazole, and posaconazole [14]. Azoles inhibit ergosterol formation in fungal cell membranes by targeting the enzyme CYP-dependent 14a-demethylase (CYP51), which converts lanosterol to ergosterol. This results in the accumulation of methylsterols, the disruption of fungal cell membranes, and the inhibition of cellular growth and replication. Adverse effects common to azoles as a class include QTc prolongation, with torsades de pointes and hepatic enzyme elevations/hepatotoxicity. Some adverse effects may be even more apparent with long-term azole exposure, such as alopecia, peripheral neuropathy, and hormone-related adverse effects. Most azoles also are known to cause gastrointestinal side effects, such as nausea and diarrhea.

The common side effects of itraconazol are gastrointestinal (abdominal pain, nausea, vomiting, and diarrhea) headache, vertigo, and skin rash. This drug should be avoided in patients with liver function abnormalities such as elevated liver enzymes and is sometimes associated with liver or biliary disorders, or even a drug-induced hepatitis. This justifies the surveillance of liver function before and after the prescription of the drug. One of the side effects is hypokalemia, particularly when the dose is >400 mg/day [24].

Voriconazole-related photopsia occurs in approximately 20% to 30% of patients; this effect seems to be related to the level of serum concentrations and typically subsides in the first week of therapy. Visual hallucinations and central nervous system (CNS) effects have also been reported with voriconazole, especially at higher serum concentrations. Skin rash, photo- sensitivity, and Stevens–Johnson syndrome have also been reported with voriconazole. An increase in squamous cell carcinomas has been observed in patients receiving voriconazole, especially with long-term exposure. Periostitis and exostoses have been reported in patients receiving voriconazole, which may be related to the accumulation of fluoride over long periods of voriconazole intake and the osteogenic impact of voriconazole on osteoblasts.

To date, adrenal insufficiency has been reported with all of these azoles, with the exception of isavuconazole; pseudohyperaldosteronism has also been reported with itraconazole and posacona-zole. Mineralocorticoid excess as well as potential direct damage to myofi- broblasts or mitochondrial dysfunction with itraconazole may lead to cardiac toxicity and congestive heart failure with its use in some patients, but this does not appear to be a class effect of azoles [25].

Griseofulvin (500–1000 mg/day) is a fungistatic with non-steroidal anti-inflamatory activity and has demonstrated the inhibition of E2 prostaglandin and vasodilator activity. The common dosage in adults is of 500–1000 mg/day, and in children, it is 10–20 mg/kg/day. The duration of the treatment is variable: 4–6 weeks in cutaneous mycosis, 6–8 weeks for tineas, and 4–6 months for onychomycosis. Griseofulvin usage is contraindicated with the intake of alcohol (antabus effect) and minimizes the action of oral contraceptives due to their inducing effect on liver enzymes. It produces interactions with oral contraceptives, cyclosporine, tacrolimus, methadone, and zidovudine. The simultaneous use of phenobarbital greatly decreases the absorption of griseofulvin. Griseofulvin is not contraindicated in children [24].

Blood count monitoring is recommended for long-term treatment with high doses (>1.5 g/day), which is rare situation nowadays. More intense surveillance in hepatic insufficiency is required (also rare situation). Its use is discouraged in patients who are pregnant or lactating.

Possible side effects are headache, dizziness, sleeplessness and sleepiness, confusion, and irritability, which are aggravated by alcohol intake (in adults). Gastrointestinal disorders and taste disturbance are also possible (in adults). Cutaneous allergic reactions and photosensitization have been noted as well. Hepatic disorders (cholestasis), hematological (anemia, leukopenia, thrombocytopenia), and a periferic neuropaty have not been observed recently [24].

Bonifaz et al. compared the efficacy and safety of terbinafine and itraconazole for the treatment of MG over a period of 18 years and concluded that both options are effective. However, terbinafine achieved a faster clinical and mycological cure and had a better safety profile than itraconazole [26].

## 3. Deep/Systemic Dermatophytosis

### 3.1. Definition

Dermatophytic disease, initially described by Hadida and Schousboe in 1959, is a chronic, rare, often serious, and sometimes life-threatening infection that is characterized by a deep and superficial invasion of the dermatophytes and their frequent dissemination to other organs. Therapeutic failure is generally reported. It is also known as systemic dermatophytosis [27].

### 3.2. Epidemiology

Dermatophytic disease is a rare disease [25], and it is primarily found in Maghreb countries, a region of North Africa that includes Morocco, Tunisia, Algeria, Libya, and Mauritania. Endogamy is frequent within these regions, which suggests autosomal recessive transmission in a context of strong inbreeding. Algeria has the highest number of cases with a reported frequency of 48.8%, followed by Morocco and Tunisia [28].

Until now, there have been 45 reported cases among individuals from North Africa. Of these, 24 patients belong to consanguineous families, 5 patients had sporadic disease, and 19 patients were from 8 different families who had the disease. The remaining 21 patients belonged to families that had not been reported to be consanguineous; of these patients, 14 had sporadic disease, and 7 had family diseases [28].

Hadida’s disease is more frequent in males, and the first symptoms appear in childhood or in early adulthood, primarily when the child is between 5 and 11 years of age [29]. However, its onset may occur when patients are anywhere between 1–50 years old (Table 1).

The most frequently found pathogens are *T. violaceum* followed by *T. rubrum* and *T. verrucosum*. Other, less frequently isolated dermatophytes are *T. schoenleinii* [29,30,31,32] and *M.canis* [33]. This is consistent with the regional epidemiology because *T. violaceum* is the most common cause of tinea capitis in North Africa, followed by *T. schoenleinii* [34,35]. The anthropophilic dermatophytes *M. audouinii*, *T. tonsurans*, and *T. soudanense* are also prevalent pathogens in Africa. Therefore, none of these species appear to be specifically associated with severe forms of dermatophytic disease [36]. Table 1 shows all of the dermatophytes that have been associated with the disease.

### 3.3. Ethiopathogeny

The pathogenesis of infectious diseases follows general principles on a spectrum: infection–disease. Infection with dermatophytic fungi is common, but diseases such as Majocchi’s granuloma is much less common, and deep/systemic dermatophytosis is rare [37].

Two significant determinants of what occurs in the infection–disease spectrum are determined by the host immune response and the pathogen’s genetic make-up. The immune response regulates whether a disease manifests or not and to what extent. For example, the CARD 9 gene mutation allows a particular disease manifestation when infected. The pathogen will also contribute to what occurs in this spectrum. Fungal genomics will be important in future reviews of this type, and it may be that specific fungal gene expressions contribute to the disease states observed in the infection-disease spectrum [37].

*CARD9* is a gene located on the long arm of chromosome 9. It encodes the intracellular adaptor protein CARD9, which plays an important role in intracellular signaling and triggers antifungal immune responses [28,38]. Mutations in the *CARD9* gene result in a premature termination codon with a subsequent abnormal production of the CARD9 protein [39].

A total of 7 different mutations have been identified in 30 patients from 15 CARD9-deficient families, which were both homozygous (Y91H, R101C, p.D274fsX60, Q289X, and Q295X) and heterozygous (G72S/R373P and p.L64fsX59/p.Q158X) and were mostly located in the CARD domain of amino acids 6–98 as well as in the coiled-coil domain of amino acids 140–420 and produced structural and functional alterations of final CARD9 protein expression [40].

Recently, autosomal recessive CARD9 deficiency has been detected in 14 patients with deep dermatophytosis and no known immunodeficiency. Most patients belonged to consanguineous families, which is why this genetic has been identified as the primary genetic cause of this deep dermatophytosis [28]. This has also been reported in immunocompromised patients with HIV infection or in patients undergoing immunosuppressive therapy [41,42]. However, antecedents of consanguinity and *CARD9* gene mutation are predominant in systemic dermatophytic disease (Table 1) [28].

A functional abnormality in cellular immunity has been evidenced in several studies that found normal CD4 and CD8 lymphocytes with the preservation of polymorphonuclear leukocyte function and increased production of IL-1 and tumor necrosis factor [43]. Type-2 CD8+ T lymphocytes also secrete IL-4 and IL-5 and are believed to be responsible for the tolerance to these dermatophytes [44].

Humoral immunity does not seem to be altered by the increase in total and specific IgE and the presence of anti-trichophyton antibodies [45].

Many analogous examples are evident in infectious disease pathogenesis literature. For example, non-toxigenic *C. diphtheriae* can cause a serious skin infection but requires toxin production to cause diphtheria in a host who lacks antibodies to the diphtheria toxin. Disseminated histoplasmosis, a rare disease, is manifested in individuals with significant CD4 T cell dysfunction/depletion who are infected with *H. capsulatum*. However, millions of individuals (with normal immune systems) in the Mississippi River and Ohio River valleys of the USA are infected with *H. capsulatum* and are symptom free. Specific genetic differences in Treponema pallidum contribute to whether patients develop neurosyphilis [37].

### 3.4. Clinical Features

Dermatophytic disease is a chronic dermatophytosis involving the skin and viscera caused by common dermatophytes crossing the skin barrier [46]. Affected family members may only present onychomycosis [47], or chronic onychomycosis, and tinea, [28] which seems to be the disease in its early stages. Of the 59 cases reviewed in the literature, 31 (52.5%) reported that their relatives, siblings, cousins, or children were affected by the condition (Table 1).

In all reported patients, the first symptoms developed during childhood or at an early age (1–51 years, average = 12.9 years) (Table 1). It initially and mainly appears as recurrent tinea of the hairy skin and/or tinea of the glabrous skin, ref. [48] although early manifestations have also been observed as extensive skin and nail dermatophytosis [40] or as only onychomycosis from an early age [49]. Later on, the disease progresses and leads to an invasive disease with adenopathies, papules or nodules, or subcutaneous nodules on the hairy skin and/or body, which may result in fistulization or ulceration, generalized desquamative erythroderma, onyxis, and alopecia of the hairy skin, eyebrows, and eyelashes [29].

There may be other manifestations that may or may not be related to the dermatophytic process, such as spleen, hepatic, pleuropulmonary, neurological and cerebral affections, peritoneal damage, and even sepsis. It is possible to observe muscle and bone damage caused by contiguity or hematogenous spread [29,50,51,52].

Consequently, there is phenotypic variability of the dermatophytic infection in CARD9 deficient patients, ranging from extensive skin and nail lesions to lymph node involvement and fatal central nervous system infection [40].

One of the characteristics of the dermatophyte fungi is that their growth appears to be restricted to the stratum corneum. Deeper penetration below the granular cell layer of the epidermis was thought to be limited by the presence of inhibitory serum factors, but later work has suggested that T-lymphocyte activation through a Th1 pathway plus neutrophil-mediated killing are the two most effective control mechanisms [53].

The nails may be a reservoir for fungus infection. The proximal subungual onychomycosis is a potential candidate for the occult but deep dissemination of dermatophytes, as there is demonstrable and histological verifiable invasion of the subungual portion of the proximal nail plate without evidence of a portal of entry either through the nail fold or by following extensive onycholysis or the undermining of the distal nail plate at its free margin [53].

### 3.5. Diagnosis

When approaching the topic of deep dermatophytic disease, research should be conducted on the intake of immunosuppressive drugs, corticosteroid use, infection by the human immunodeficiency virus, and lymphopenia. If none of these risk factors are identified, CARD9 deficiency should be investigated, especially in North African patients with clinical signs that begin during childhood [40].

The histopathology of subcutaneous nodules shows tuberculoid granulomas that present, from the periphery to the center, an area of fibrosis, epithelioid cells, multinucleated giant cells, and plasma cells, with necrotic deposits in the center and that are surrounded by polymorphonuclear cells. PAS-positive filamentous structures can be seen within the area of necrosis [29]. Immunohistochemical tests with primary anti-dermatophyte monoclonal antibodies yield positive results [28]. Histological examination of the lymph nodes reveals granulomas containing hyphae and necrosis. A culture of the affected tissue is necessary to confirm the causative agent [28].

Intradermal reaction in these patients has been observed to be predominantly positive for tuberculin and has shown a lack of response to trichophyton (Table 1).

Normal CD4+ and CD8+ T-lymphocyte, B-lymphocyte, and NK-lymphocyte subsets can be observed [28]. The lymphocyte transformation test, or transformation test, specifically measures the proliferation of memory T-lymphocytes in the presence of a specific antigen and the expression of the markers on its surface. It is useful for detecting the immune functional capacity of cell-mediated reactions [54].

With polymerase chain reaction (PCR) as well as with dermatophyte-PCR-ELISA, fungi can be detected in clinical material directly in a highly specific and sensitive manner without prior culture. Currently, molecular methods such as matrix-assisted laser desorption/ionization time-of-flight mass spectrometry (MALDI TOF–MS) as culture confirmation assays complete conventional mycological diagnostics [55].

**Table 1 jof-07-00929-t001:** Description of patients with deep/systemic dermatophytosis reported in the literature.

Case No.	Age at Onset (Years)	Age (Years)	Sex	Country of Origin	Consanguinity/ Family Member Affected	Organs Affected	Treatments	Dx/Agent	Special Tests
1 [56]	NR	NR	M	Algeria		Skin		*T. faviforme*	
2 [27]	NR	11	M	Algeria	Yes/-	NR			
3 [27]	NR	13	M	Algeria	NS/Brother	NR			
4 [25,48]	NR	NR	NR	Morocco		Skin, hairy skin, lymph nodes		*T. violaceum*	
5 [57]	NR	NR	NR	Morocco		Skin		*T. violaceum*	
6 [58]	NR	18	M	Algeria		Skin, nails		*T. verrucosum*	
7 [58]	NR	26	M	Algeria	NS/Brother	Skin nails		*T. verrucosum*	
8 [31]	14	22	F	Algeria	NS/Siblings	Skin involvement		*T. schoenleinii*	
9 [31]	26	35	M	Algeria	NR	NR		*T. verrucosum*	
10 [25,59]	NR	NR	NR		NS/Cousin	NR			
11 [30]	NR	35	M	Tunisia	NR	Skin, nails		*T. violaceum* *T. schoenleinii*	
12 [52]	NR	43	M	Japan	NR	Onyxis	NR	*T. mentagrophytes var. interdigitale*	NR
13 [59]	6	11	M	Algeria	Yes/-	Tinea, Skin involvement, onyxis		*T. violaceum*	
14 [59]	11	21	M	Algeria	Yes/Brother	Tinea, Skin involvement, onyxis			
15 [59]		4	M	Algeria	Yes/Brother	Tinea			
16 [59]	15	41	M	Algeria	Yes/Brother	Tinea, Skin involvement, onyxis		*T. verrucosum*	
17 [59]	6	25	M	Algeria	Yes/Brother	Tinea, Skin involvement, onyxis		*T. verrucosum*	
18 [60]	6	42	M	Tunisia		Skin, nails			
19 [25,61]								*T. violaceum*	
20 [33]	9	40	F	Algeria	Yes/-	Skin, hairy skin, nails, lymph nodes	Griseofulvin	Culture/*T. rubrum*	M.CARD9 Q289X/Q289X
21 [25,62]									
22 [43]	6	53	M	Algeria	Yes/Cousin	Skin involvement, hairy skin, nails, lymph nodes	Griseofulvin Econazole itraconazole	Culture/ *T. violaceum*	M.CARD9 Q289X/Q289X
23 [49]	2	28	M	Algeria	NS/Cousin	Skin, hairy skin, nails, lymph nodes, brain	Griseofulvin Ketoconazole Itraconazole	Culture/ *T. violaceum*	NR
24 [48]	5	15	M	Morocco	NS/Brother	Skin, hairy skin, nails, lymph nodes, hypogonadism	Griseofulvin	Culture/ *T. violaceum*	IDR Tb +
25 [48]	6	11	M	Morocco	NS/Brother	Skin, hairy skin, nails, lymph nodes, hypogonadism	Ketoconazole, Griseofulvin	*T. tonsurans*	IDR TF + IDR CD + IDR Tb +
26 [48]	22	23	M	Morocco	NR	Skin, hairy skin, nails	Griseofulvin	Culture/ *T. violaceum*	NR
27 [48]	22	27	M	Morocco	NS/Cousin	Skin, nails		T. violaceum	IDR Tb +
28 [48]	NR	NR	M	Morocco	NS/Cousin	Skin, nails, lymph nodes, death	Griseofulvin	Culture/ *T. violaceum*	IDR Tb +
29 [48]	10	20	M	Morocco	NS/Cousin	Skin, hairy skin, nails, lymph nodes, lung, deterioration of general condition	Griseofulvin	Culture/ *T. violaceum*	IDR CD + IDR Tb +
30 [48]	29	30	F	Morocco	NR	Skin, nails, lymph nodes, death	Griseofulvin	Culture/ *T. violaceum*	IDR Tb +
31 [48]	10	25	F	Morocco	NR	Skin, hairy skin, nails	Griseofulvin Ketoconazole	Culture/ *T. violaceum*	IDR TF − IDR CD − IDR Tb −
32 [44]	8	34	M	Algeria	Yes/3 siblings	Skin, hairy skin, nails, death	NR	Culture/ *T. violaceum*	NR
33 [63]	4	42	NR	Algeria	NS/Brother	Tinea, skin involvement, onyxis		*T. violaceum*	
34 [64]	NR	29	M	Algeria	Yes/-	Skin, hairy skin, nails, lymph nodes, death	Griseofulvin ketoconazole topical	Culture/ *T. violaceum*	IDR TF −, IDR CD −, IDR Tb + M.CARD9 Q289X/Q289X
35 [65]	NR	NR	NR	NR	NR	NR	NR	T. violaceum *T. rubrum*	NR
36 [66]	50	60	M	NR	NR	Skin, nails, lymph nodes	Griseofulvin itraconazole fluconazole terbinafine	T. violaceum *T. rubrum*	NR
37 [67]	18	23	M	NR	NR	Skin, hairy skin, nails, lymph nodes	Itraconazole	*T. verrucosum*	IDR Tb -
38 [29]	1	8	F	Tunisia	Yes/Brother	Skin, lymph nodes, nails, hairy skin	Griseofulvin Itraconazole Ketoconazole Terbinafine Surgical resection	Culture/ *M. canis* *T. violaceum*	IDR TF − IDR CD − IDR Tb +
39 [29]	1	7	F	Tunisia	Yes/Brother	Skin, hairy skin, nails	Griseofulvin Itraconazole Topical antifungals Surgical resection	Culture/ *M. canis* *T. violaceum*	IDR TF − IDR CD − IDR Tb +
40 [29]	1	5	F	Tunisia	Yes/Brother	Skin, hairy skin, nails, lymph nodes	Griseofulvin Itraconazole Ketoconazole Fluconazole Topical antifungals Antibiotics Surgical resection	Culture/ *M. canis* *T. violaceum*	IDR TF − IDR CD − IDR Tb +
41 [47]	18	38	M	Algeria	NS/1 brother with onychomycosis of hands and feet	Skin, hairy skin, nails	Griseofulvin Fluconazole	NR	IDR Tb +
42 [47]	NR	NR	M	Algeria	NS/2 siblings	Skin, hairy skin, nails	Griseofulvin	*Filaments*	IDR Tb −
43 [32]	NR	NR	NR	Algeria	NR	Skin, hairy skin, nails, death	Griseofulvin	*T. schoenleinii*	IDR TF −, IDR Tb +
44 [32]	NR	NR	NR	Algeria	NR	Skin, hairy skin, nails, death	NR	*T. violaceum*	IDR TF −, IDR Tb +
45 [68]	16	52	M	Algeria	NR	Skin, hairy skin, nails, lymph nodes, brain	Ketoconazole Fluconazole Cefotaxime Amikacin Phenobarbital Metronidazole	*Filaments*	IDR Tb −
46 [69]	3	27	F	Tunisia	Yes/-	Skin, nails, lymph nodes	Itraconazole	*T. violaceum*	NR
47 [28]	8	56	M	Algeria	Yes/Sibling	Skin, hairy skin, nails	Griseofulvin	Culture/ *T. violaceum*	M.CARD9 Q289X/Q289X
48 [28]	19	43	M	Algeria	Yes/Sibling	Skin, hairy skin, nails, lymph nodes	Griseofulvin Fluconazole	Biopsies with fungal hyphae	M.CARD9 Q289X/Q289X
49 [28]	21	40	M	Algeria	Yes/Sibling	Skin, hairy skin, gastrointestinal (anal stenosis)	Griseofulvin Fluconazole Colostomy surgery Terbinafine	Biopsies with fungal hyphae	M.CARD9 Q289X/Q289X
50 [28]	NR	28	M	Algeria	Yes/Sibling	Skin, hairy skin, death	NR	Biopsies with fungal hyphae	NR
51 [28]	NR	40	M	Morocco	Yes/Sister	Skin, hairy skin, nails, lymph nodes, bone (Osteolysis of 1st and 2nd left toes)	Terbinafine Voriconazole Posaconazole Liposomal amphotericin B Interferon-y Amputation of 2nd left toe	Culture/ *T. rubrum*	M.CARD9 R101C/R101C
52 [28]	NR	49	F	Morocco	Yes/Brother	Skin, hairy skin, nails	NR	NR	M.CARD9 R101C/R101C
53 [28]	6	91	M	Tunisia	NS/Offspring	Skin, hairy skin, nails, death	NR	NR	M.CARD9 Q289X/Q289X
54 [28]	12	44	M	Tunisia	NS/Sister	Skin, nails	Fluconazole Itraconazole	Culture/ *T. rubrum*	M.CARD9 Q289X/Q289X
55 [28]	5	52	F	Tunisia	NS/Sibling	Skin, hairy skin, nails, lymph nodes	Griseofulvin Ketoconazole Fluconazole	*T. violaceum* *T. rubrum*	M.CARD9 Q289X/Q289X
56 [28]	6	62	M	Tunisia	NR	Skin, hairy skin, nails, lymph nodes	Griseofulvin Fluconazole Terbinafine Itraconazole Voriconazole	*T. rubrum* *T. violaceum*	M.CARD9 Q289X/Q289X
57 [28]	8	41	F	Algeria	Yes/Siblings	Nails *	Griseofulvin	*T. violaceum*	M. CARD9 Q289X/Q289X
58 [28]	NR	37	F	Algeria	Yes/Brother	Nails **		Dermatophyte	M. CARD9 Q289X/Q289X
59 [40]	13	4th decade	M	Gharbia, Egypt	NR	Skin, nails	Itraconazole Ketoconazole Terbinafine Ketoconazole Topical Cyclopyroxolamine Posaconazole	Culture/ *T. rubrum*	M.CARD9 Q289X/Q289X
60 [69]	51	65	M	NR	NR	Skin, hairy skin, nails, lymph nodes	Griseofulvin	Culture/ *T. violaceum*	NR
61 [70]	10	47	F	Algeria	Yes/-	Skin, hairy skin, lymph nodes, brain	Griseofulvin Itraconazole	Culture/ *T. rubrum*	M.CARD9 Q289X/Q289X
62 [71]	16	31	M	NR	NR	Skin, nails, lymph nodes	Griseofulvin Ketoconazole Itraconazole Posaconazole Amphotericin B	T. rubrum *T. violaceum*	M.CARD9 intron 3: c.271T > C (p.Y91H) Intron 8: c.1269 + 18G > A
63 [72]	3	24	M	Brazil	Not	Skin, hairy skin, nails	Nistatin Ketoconazole Itraconazole Terbinafin Posaconazole Amphotericin B	Culture/*T. mentagrophytes*	Homozygous for a novel c.302G > T variation in the exon 3 of the CARD9 gene (R101L)

NR: not reported, F: female, M: male, NS: not specified, TF: trichophyton, CD: candidin, Tb: tuberculin, +: positive, −: negative, M.CARD9: CARD9 mutation; * onychomycosis of all nails; ** chronic onychomycosis since childhood.

### 3.6. Differential Diagnosis

Differential diagnosis is performed with deep dermatophytosis and extensive dermatophytosis. Immunosuppression secondary to immunosuppressive therapy, corticosteroids, or human immunodeficiency virus infection should be deliberately sought [40].

MG is generally limited to a single or multiple perifollicular areas [42].

### 3.7. Treatment

Griseofulvin therapy is the most widely used treatment for dermatophytic disease, followed by itraconazole. However, severe relapses are reported after the discontinuation of antifungal therapy. Therefore, treatment with antifungals must be chronic, even for several years, and can sometimes be associated with oral antibiotics or the surgical resection of the nodule and bulkier lymphadenopathy [29]. Some patients require lifelong antifungal maintenance therapy [28]. Jachiet et al. reported a case of a patient with recurrent extensive erythematous lesions and onychomycosis who had received multiple antifungals until posaconazole was finally administered, which produced a good response and led to the achievement of sustained clinical remission, unlike other antifungals with antidermatophytic activity. However, this was in a single-patient case [40].

Prophylactic antifungal therapy has been proposed to limit disease progression as long with genetic counseling for families [46].

## Figures and Tables

**Figure 1 jof-07-00929-f001:**
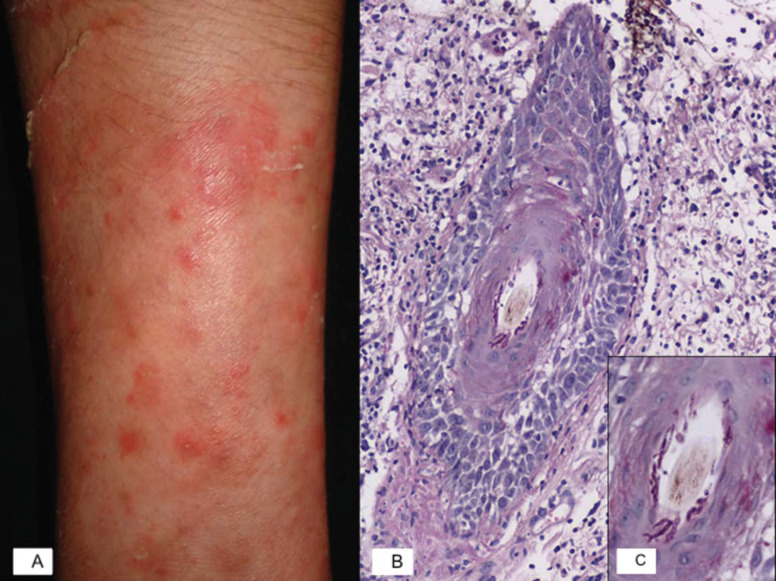
Inflammatory tinea: Majocchi’s granuloma. (**A**) Majocchi granuloma characterized by erythematous nodules on the forearm. (**B**) Microphotography of a PAS-stained slide shows a vellus hair with adjacent mixed inflammatory infiltrate, epithelium spongiosis, exocytosis of neutrophils, and hyphae surrounding the hair shaft. (**C**) Closer view of the vellus hair showing septate hyphae within the follicular canal.

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
