# Peer review of "Unusual Inflammatory Tinea Infections: Majocchi’s Granuloma and Deep/Systemic Dermatophytosis"

_jof, 2021, doi:10.3390/jof7110929_

Round 1
Reviewer 1 Report
The second part of the title should be edited to change "Dermatophytic or Hadida’s Disease" to "Deep/systemic dermatophytosis". There is superficial clinical resemblance between Majocchi's granuloma and deep/systemic dermatophytosis. The pathogenesis of the latter condition is related to defects in the immune response that permit invasive fungal disease by fungal pathogens that remain in superficial locations in patients with normal immune responses. The authors should relate the two topics of their review in the introduction. Reference to Hadida/Schousboe should remain in the text.
Treatment of each syndrome requires prolonged treatment with agents with significant side effects. The authors should discuss side effects of the major therapeutic agents/monitoring, even if briefly stated.
Pathogenesis of infectious diseases follows general principles with the spectrum: infection <--> disease. Infection with dermatophytic fungi is common but disease such as Majocchi’s granuloma is much less common and deep/systemic dermatophytosis is rare.
Two significant determinants of what occurs in the spectrum of infection <--> disease are determined by the host immune response and the pathogen's genetic make-up (see Stanley Falkow, Nat Rev Microbiol. 2004 Jan;2(1):67-72. doi:10.1038/nrmicro799). The immune response regulates whether disease is manifest or not and to what extent. For example, the CARD 9 gene mutation allows a particular disease manifestation when infected. The pathogen will also contribute to what occurs in this spectrum. Fungal genomics will be important in future reviews of this type and it may be that specific fungal gene expressions contribute to the disease states observed in the infection <--> disease spectrum.
Many analogous examples are evident in infectious diseases pathogenesis literature. For example, non-toxigenic C. diphtheriae can cause serious skin infection but require toxin production to cause diphtheria in a host who lacks antibodies to diphtheria toxin. Disseminated histoplasmosis, a rare disease, is manifested in individuals with significant CD4 T cell dysfunction/depletion who are infected with H. capsulatum. However, millions of individuals (with normal immune systems) in the Mississippi River and Ohio River valleys of the U.S.A. are infected with H. capsulatum and are symptom free. Specific genetic differences in Treponema pallidum contribute to whether patients develop neurosyphilis.
A brief discussion of this concept would contribute to the manuscript.
Author Response
Point 1. The second part of the title should be edited to change "Dermatophytic or Hadida’s Disease" to "Deep/systemic dermatophytosis". There is superficial clinical resemblance between Majocchi's granuloma and deep/systemic dermatophytosis. The pathogenesis of the latter condition is related to defects in the immune response that permit invasive fungal disease by fungal pathogens that remain in superficial locations in patients with normal immune responses. The authors should relate the two topics of their review in the introduction. Reference to Hadida/Schousboe should remain in the text.
Response 1.
The second part of the title is changed to: "Deep/systemic dermatophytosis".
Dermatophytes are fungi that invade and multiply within keratinized tissues (skin, hair, and nails) causing infection. The new phylogenetic classification of dermatophytes includes 9 genre, and those that affect humans are Trichophyton, Microsporum, Epidermophyton and Nannizzia. Finally, based upon the affected site, these have been classified clinically into tinea capitis, tinea faciei, tinea barbae, tinea corporis, tinea manus, tinea cruris, tinea pedis, and tinea unguium .
Other unusual clinical variants include Majocchi granuloma and Dermatophytic disease; although usually painless and superficial, these fungi can behave in an invasive manner, causing deeper and disseminated infection and should not be neglected.
Majocchi's granuloma represents a rare, deep dermatophyte folliculitis, it is considered a clinical variant of tinea corporis usually caused by the dermatophyte T. rubrum. Two main clinical forms have been described: the papulopustular perifollicular and indurated plaques with erythematous subcutaneous nodules. The former is typically developed in trauma-prone areas in healthy individuals, whereas the latter presents in immunosuppressed patients.
Dermatophytic disease is a chronic dermatophyte infection of the skin and viscera, occurring mainly in Africa. Immunological studies have highlighted a deficit of cellular immunity with autosomal recessive transmission responsible for tolerance to dermatophyte. The first signs of this disease usually occur during childhood; is a serious life-threatening disease due to its inexorable evolution toward visceral involvement.
Dermatophytic disease is a chronic dermatophyte infection of the skin and viscera, occurring mainly in Africa. Immunological studies have highlighted a deficit of cellular immunity with autosomal recessive transmission responsible for tolerance to dermatophyte. The first signs of this disease usually occur during childhood; is a serious life-threatening disease due to its inexorable evolution toward visceral involvement. The present review is an important topic and will explore the recent advances in the pathophysiology and management.
Point 2. Treatment of each syndrome requires prolonged treatment with agents with significant side effects. The authors should discuss side effects of the major therapeutic agents/monitoring, even if briefly stated.
Response 2.
Terbinafine
Terbinafine has a fungistatic action by the inhibiton of ergosterol synthesis (by the blockade of squalene epoxidase) and fungicidal, particularly against dermatophytes (by accumulation of squalene in a proximal step before squalene epoxidase)
Gastrointestinal side effects are frequent (5%) such as dyspepsia, retching, diarrhea and abdominal pain; and taste disturbances (2%) like dysgeusia and also reversible taste loss that can last even 2 months. It's described exanthema and urticaria, appetite loss, headache and visual disturbances (blurry vision, visual deficiency and decreased visual acuity) arthralgias, myalgias, depression and fatigue. Serious side effects are mainly dermatological; exacerbation of eczema, of psoriasis, or lupus erythematosus, and more serious; acute exanthematic pustulosis, drug reaction with esosinofilia, and systemic symptoms, Stevens-Jhonson syndrome and toxic epidermal necrolysis.
A study of liver function is recommended before prescribing terbinafine and repeat it 4-6 weeks after using the drug or if the clinical symptomatology suggests liver involvement. An elevation of liver enzymes justifies the immediate interruption of terbinafine. However this phenomenon is rare. Severe liver failure has been described, in most cases, in patients with a severe underlying condition. Some cases of neutropenia, thrombocytopenia, agranulocytosis and pancytopenia reported during the use of terbinafine may justify a monitoring of the complete blood count.
Itraconazole, voriconazole y posaconazole
Azoles (itraconazole, voriconazole, posaconazole) inhibit ergosterol formation in fungal cell membranes by targeting the enzyme CYP-dependent 14a-demethylase (CYP51), which converts lanosterol to ergosterol. This results in accumulation of methylsterols, disruption of fungal cell membranes, and inhibition of cellular growth and replication.
Adverse effects common to azoles as a class include QTc prolongation, with torsades de pointes and hepatic enzyme elevations/hepatotoxicity. Some adverse effects may be even more apparent with long-term azole exposure, such as alopecia, peripheral neuropathy, and hormone-related adverse effects. Most azoles also are known to cause gastrointestinal side effects, such as nausea and diarrhea.
The common side effects of itraconazol are gastrointestinal (abdominal pain, nausea, vomiting, and diarrhea) headache, vertigo, skin rash. This drug should be avoided in patients with liver function abnormalities such as elevated liver enzymes, sometimos associated with liver or biliary disorders or even a drug-induced hepatitis. This justifies the surveillance of liver function before and after the prescription of the drug. One of the side effects is hypokalemia, particularly when the dose is >400 mg/day.
Voriconazole-related photopsia occurs in approximately 20% to 30% of patients; this effect seems related to the level of serum concentrations and typically subsides in the first week of therapy. Visual hallucinations and central nervous system (CNS) effects also have been reported with voriconazole, especially at higher serum concentrations. Skin rash, photo- sensitivity, and Stevens-Johnson syndrome also have been reported with voriconazole. An increase in squamous cell carcinomas has been observed in patients receiving voriconazole, especially with long-term exposure. Periostitis and exostoses have been reported in patients receiving voriconazole and may relate to accumulation of fluoride over long periods of voriconazole intake and osteogenic impact of voriconazole on osteoblasts.
Adrenal insufficiency has been reported with all of these azoles with the exception of isavuconazole to date; pseudohyperaldosteronism also has been reported with itraconazole and posacona- zole. Mineralocorticoid excess as well as potential direct damage to myofi- broblasts or mitochondrial dysfunction with itraconazole may lead to cardiac toxicity and congestive heart failure with its use in some patients, but this does not appear to be a azole class effect.
Griseofulvin
Griseofulvin is a fungistatic with non-steroidal anti-inflamatory activity, inhibition of E2 prostaglandin and vasodilator activity. Dosage in adults is of 500-1,000 mg/day, in children 10-20 mg/kg/day. The duration of the treatment is variable, of 4-6 weeks in cutaneous mycosis, 6-8 weeks for tineas, and 4-6 months for onychomycosis. Griseofulvin usage is contraindicated with the intake of alcohol (antabus effect) and minimizes the action of oral contraceptives due to their inducing effect on liver enzymes. It produces interactions with oral contraceptives, cyclosporine, tacrolimus, methadone, and zidovudine. Simultaneous use of phenobarbital, greatly decreases the absorption of griseofulvin. Griseofulvin is not contraindicated in children.
Monitoring of the blood count is recommended in a long duration treatment with high doses (>1.5 gr/day) rare situation nowadays. More intense surveillance in hepatic insufficiency (also rare situation). Its use is discouraged in pregnancy and lactation.
Possible side effects are headache, dizziness, sleeplessness and sleepiness, confusion and irritability are empowered by alcohol intake (in adults). Gastrointestinal disorders, taste disturbance (in adults). Cutaneous allergic reactions and photosensitization. Hepatic disorders (cholestasis), hematological (anemia, leukopenia, thrombocytopenia) and a periferic neuropaty are not observed currently.
Point 3. Pathogenesis of infectious diseases follows general principles with the spectrum: infection <--> disease. Infection with dermatophytic fungi is common but disease such as Majocchi’s granuloma is much less common and deep/systemic dermatophytosis is rare.
Two significant determinants of what occurs in the spectrum of infection <--> disease are determined by the host immune response and the pathogen's genetic make-up (see Stanley Falkow, Nat Rev Microbiol. 2004 Jan;2(1):67-72. doi:10.1038/nrmicro799). The immune response regulates whether disease is manifest or not and to what extent. For example, the CARD 9 gene mutation allows a particular disease manifestation when infected. The pathogen will also contribute to what occurs in this spectrum. Fungal genomics will be important in future reviews of this type and it may be that specific fungal gene expressions contribute to the disease states observed in the infection <--> disease spectrum.
Many analogous examples are evident in infectious diseases pathogenesis literature. For example, non-toxigenic C. diphtheriae can cause serious skin infection but require toxin production to cause diphtheria in a host who lacks antibodies to diphtheria toxin. Disseminated histoplasmosis, a rare disease, is manifested in individuals with significant CD4 T cell dysfunction/depletion who are infected with H. capsulatum. However, millions of individuals (with normal immune systems) in the Mississippi River and Ohio River valleys of the U.S.A. are infected with H. capsulatum and are symptom free. Specific genetic differences in Treponema pallidum contribute to whether patients develop neurosyphilis.
A brief discussion of this concept would contribute to the manuscript.
Response 3.
Pathogenesis of infectious diseases follows general principles with the spectrum: infection - disease. Infection with dermatophytic fungi is common but disease such as Majocchi’s granuloma is much less common and deep/systemic dermatophytosis is rare.
Two significant determinants of what occurs in the spectrum of infection - disease are determined by the host immune response and the pathogen's genetic make-up. The immune response regulates whether disease is manifest or not and to what extent. For example, the CARD 9 gene mutation allows a particular disease manifestation when infected. The pathogen will also contribute to what occurs in this spectrum. Fungal genomics will be important in future reviews of this type and it may be that specific fungal gene expressions contribute to the disease states observed in the infection - disease spectrum.
Many analogous examples are evident in infectious diseases pathogenesis literature. For example, non-toxigenic C. diphtheriae can cause serious skin infection but require toxin production to cause diphtheria in a host who lacks antibodies to diphtheria toxin. Disseminated histoplasmosis, a rare disease, is manifested in individuals with significant CD4 T cell dysfunction/depletion who are infected with H. capsulatum. However, millions of individuals (with normal immune systems) in the Mississippi River and Ohio River valleys of the U.S.A. are infected with H. capsulatum and are symptom free. Specific genetic differences in Treponema pallidum contribute to whether patients develop neurosyphilis.
Reviewer 2 Report
Unusual inflammatory tinea infections
- Please make introduction part, and add the reason you select only 2 diseases, Majocchi’s granuloma and dermatophytic disease.
- In table 1, almost cases are from Africa. The patients from other continent were not included. Why?
- Please comment about the relation dermatophytic disease with other dermatophytosis like systemic dermatophytosis, severe dermatophytosis, deep dermatophytosis, extensive dermatophytosis, tine profunda. Why don’t you make new proper name of the dermatophytic disease? Do you think CARD 9 deficency is main cause of dermatophytic disease?
- Figure
Please show more granuloma in Figure 1B.
Clinical pictures of dermatophytic disease will help for readers.
5.Error
Line 2 Tines -> Tinea
Line 72 herpes -> plaque(?)
Line 100 (Tinea cruris) -> (tinea cruris)
Line 147 hadida -> Hadida
Line 265 Jachiet M et al -> Jachiet et al
Line 427 Ebongo Aboutou CN, Soumya Chihab FH -> Aboutou CN, Hali F, Chihab S
Trichophyton and Microsporum can be replaced with T. and M.

Author Response
Point 1. Please make introduction part, and add the reason you select only 2 diseases, Majocchi’s granuloma and dermatophytic disease.
Response 1.
Dermatophytes are fungi that invade and multiply within keratinized tissues (skin, hair, and nails) causing infection. The new phylogenetic classification of dermatophytes includes 9 genre, and those that affect humans are Trichophyton, Microsporum, Epidermophyton and Nannizzia. Finally, based upon the affected site, these have been classified clinically into tinea capitis, tinea faciei, tinea barbae, tinea corporis, tinea manus, tinea cruris, tinea pedis, and tinea unguium .
Other unusual clinical variants include Majocchi granuloma and Dermatophytic disease; although usually painless and superficial, these fungi can behave in an invasive manner, causing deeper and disseminated infection and should not be neglected.
Majocchi's granuloma represents a rare, deep dermatophyte folliculitis, it is considered a clinical variant of tinea corporis usually caused by the dermatophyte T. rubrum. Two main clinical forms have been described: the papulopustular perifollicular and indurated plaques with erythematous subcutaneous nodules. The former is typically developed in trauma-prone areas in healthy individuals, whereas the latter presents in immunosuppressed patients.
Dermatophytic disease is a chronic dermatophyte infection of the skin and viscera, occurring mainly in Africa. Immunological studies have highlighted a deficit of cellular immunity with autosomal recessive transmission responsible for tolerance to dermatophyte. The first signs of this disease usually occur during childhood; is a serious life-threatening disease due to its inexorable evolution toward visceral involvement. The present review is an important topic and will explore the recent advances in the pathophysiology and management.
Point 2. In table 1, almost cases are from Africa. The patients from other continent were not included. Why?
Response 2. An exhaustive review was done of the reported cases of Hadida’s disease, including all continents, however it is in Africa where most of the reported cases are. We added a Brazilian patient.
Point 3. Please comment about the relation dermatophytic disease with other dermatophytosis like systemic dermatophytosis, severe dermatophytosis, deep dermatophytosis, extensive dermatophytosis, tine profunda. Why don’t you make new proper name of the dermatophytic disease? Do you think CARD 9 deficency is main cause of dermatophytic disease?
Response 3.
Also known as dermatophytic disease, systemic dermatophytosis, severe dermatophytosis, deep dermatophytosis, extensive dermatophytosis, is characterized by dermis and hypodermis invasion, sometimes affecting lymph nodes, digestive tract, brain and bones.
Affected patients by deep dermatophytosis are mainly of north Africa having in common the presence of the mutation of the gene that codifies for the CARD 9 protein. Therefore we propose that this disease has the name of dermatophytosis by CARD 9 deficiency.
Point 4. Figure
Please show more granuloma in Figure 1B.
Clinical pictures of dermatophytic disease will help for readers.
Response 4.
Authors do not count with clinical images of dermatophytic disease
Point 5. Error
Line 2 Tines -> Tinea
Line 72 herpes -> plaque(?)
Line 100 (Tinea cruris) -> (tinea cruris)
Line 147 hadida -> Hadida
Line 265 Jachiet M et al -> Jachiet et al
Line 427 Ebongo Aboutou CN, Soumya Chihab FH -> Aboutou CN, Hali F, Chihab S
Trichophyton and Microsporum can be replaced with T. and M.
Response 5.
Line 2 Tines -> Tinea
Line 72 herpes -> plaques
Line 100 (Tinea cruris) -> tinea cruris
Line 147 hadida -> Hadida
Line 265 Jachiet M et al -> Jachiet et al
Line 427 Ebongo Aboutou CN, Soumya Chihab FH -> Aboutou CN, Hali F, Chihab S
Trichophyton and Microsporum is replaced with T. and M.
Reviewer 3 Report
Title-Tinea, not Tines
Maladie dermatophytique is the term used by Hadida and Schousboe and is a good descriptive term and roughly translates to Dermatophytic Disease. The term ,Dermatophytic disease ,is periodically used in the literature, but may be misleading to many readers as it is literally a general term. Lines 155 and 206-, suggest using a different term than Dermatophytic.
Lines 42-46- Uncomplicated Tinea capitis in the USA is more common before age of puberty. Make this clear.
Reference Robert Baran's papers on nails and deeper dermatophyte infection. Also mention to check the nails since they may be a reservoir for fungus infection.
Shorten the text of the paper
Author Response
Point 1. Title-Tinea, not Tines
Response 1. Title - Tinea
Point 2. Maladie dermatophytique is the term used by Hadida and Schousboe and is a good descriptive term and roughly translates to Dermatophytic Disease. The term ,Dermatophytic disease ,is periodically used in the literature, but may be misleading to many readers as it is literally a general term. Lines 155 and 206-, suggest using a different term than Dermatophytic.
Response 2. Dermatophytic disease is replaced with deep dermatophytic
Point 3. Lines 42-46- Uncomplicated Tinea capitis in the USA is more common before age of puberty. Make this clear.
Response 3. The disease is of cosmopolitan nature, predominantly occurring in the age groups between 20 and 35 years; generalized MG cases have been described in children aged 3–5 years. Uncomplicated tinea capitis is more common before age of puberty, however, when it occurs after this age it is more common in women. MG is predominant among females in a 3:1 ratio because they are more vulnerable to developing tinea capitis after puberty and they generally shave their legs. MG cases 45 among men are associated with immunosuppression.
Point 4. Reference Robert Baran's papers on nails and deeper dermatophyte infection. Also mention to check the nails since they may be a reservoir for fungus infection.
Response 4.
One of the characteristics of the dermatophyte fungi is that their growth appears to be restricted to the stratum corneum. Deeper penetration below the granular cell layer of the epidermis was thought to be limited by the presence of inhibitory serum factors, but later work has suggested that T-lymphocyte activation through a Th1 pathway plus neutrophil-mediated killing are the two most effective control mechanisms.
The nails may be a reservoir for fungus infection. The proximal subungual onychomycosis is a potential candidate for occult but deep dissemination of dermatophytes as there is demonstrable and histological verifiable invasion of the subungual portion of the proximal nail plate without evidence of a portal of entry either through the nail fold or following extensive onycholysis, or undermining of the distal nail plate at its free margin.
Round 2
Reviewer 3 Report
Thank you for making the changes